# Probing the Effects of the FGFR-Inhibitor Derazantinib on Vascular Development in Zebrafish Embryos

**DOI:** 10.3390/ph14010025

**Published:** 2020-12-30

**Authors:** Maria P. Kotini, Felix Bachmann, Jochen Spickermann, Paul M. McSheehy, Markus Affolter

**Affiliations:** 1Biozentrum der Universität Basel, Klingelbergstrasse 50/70, CH-4053 Basel, Switzerland; maria.kotini@unibas.ch; 2Basilea Pharmaceutica International Ltd., Grenzacherstrasse 487, CH-4005 Basel, Switzerland; felix.bachmann@basilea.com (F.B.); jochen.spickermann@basilea.com (J.S.); paul.mcsheehy@basilea.com (P.M.M.)

**Keywords:** angiogenesis, anastomosis, vascular development, derazantinib, FGFR-signalling, VEGFR-signalling, infigratinib, vatalanib, zebrafish

## Abstract

Angiogenesis is a fundamental developmental process and a hallmark of cancer progression. Receptor tyrosine kinases (RTK) are targets for cancer therapy which may include their action as anti-angiogenic agents. Derazantinib (DZB) is an inhibitor of the fibroblast growth factor receptors (FGFRs) 1–3 as well as other kinase targets including vascular endothelial growth factor receptor 2 (VEGFR2), colony stimulating factor-1 receptor (CSF1R) and platelet-derived growth factor beta receptor (PDGFRbeta). This study aimed to investigate the effect of DZB on blood vessel morphogenesis and to compare its activity to known specific FGFR and VEGFR inhibitors. For this purpose, we used the developing vasculature in the zebrafish embryo as a model system for angiogenesis in vivo. We show that DZB interferes with multiple angiogenic processes that are linked to FGF and VEGF signalling, revealing a potential dual role for DZB as a potent anti-angiogenic treatment.

## 1. Introduction

Angiogenesis is a physiological process in which new blood vessels are formed from pre-existing vessels [1]. Angiogenesis occurs during normal growth and development, and under physiological conditions, and is tightly regulated by the coordinated actions of both anti-angiogenic [2] and pro-angiogenic factors, of which the latter include vascular endothelial growth factor (VEGF), platelet-derived growth factor (PDGF) and fibroblast growth factor (FGF) and their respective receptors [3]. Tumor angiogenesis is one of the hallmarks of cancer and plays a crucial role in providing oxygen and nutrients to tumor cells during cancer progression and metastasis. Thus, disruption of this process has been a major area of research which has led to a number of useful clinical treatments including bevacizumab, aflibercept, ramucirumab and small-molecule VEGFR inhibitors such as sunitinib and vatalanib [3,4]. Studies of bevacizumab and sunitinib in renal cell carcinoma have identified a potential angiogenic-signature of response consisting of seven genes—VEGFR1 and 2, VEGFA and four different endothelial markers [5]. The importance of immune-infiltrating cells such as T-cells and macrophages has also been recognized, which has encouraged combinations of immune checkpoint blockade combined with VEGFR2 inhibitors [6,7]. Interestingly, the most effective combination may be when a relatively weak anti-angiogenic effect is employed since this results in a greater immune infiltration [8].

Derazantinib (DZB) is an oral small molecule inhibitor of FGFR1–3, with similar potency against CSF1R, a cell-surface receptor important for regulating the activity of M2-macrophages [9,10]. DZB has shown significant efficacy in patients with cholangiocarcinoma in a phase-1/2 trial [11] and further trials have been initiated in both urothelial (NCT04045613) and gastric cancer (NCT04604132). A biochemical kinase-screen indicated that other kinases may be targeted by DZB including VEGFR2 [9], so we hypothesized that DZB may have activity in the zebrafish vasculature, a model which is recognized as a useful in vivo assay for studying the mechanism of action of putative anti-angiogenic agents [12,13]. The developing zebrafish vasculature is a well-established system for the investigation of blood vessel formation and maintenance [14]. VEGF receptors and FGF receptors are expressed in this system [13,15] and the requirement of VEGF and FGF pathways in vascular development has been shown in various systems including the those of the zebrafish [13,16,17].

In this study, we investigated the activity of the small-molecule kinase inhibitor DZB (BAL087) in the vascular development of zebrafish (*Danio rerio*) and compared it with vatalanib (aka, PTK/ZK), which is a specific inhibitor of VEGFR1–3, and with infigratinib (also known as BGJ398), which is a specific inhibitor of FGFR1–3 [18,19].

## 2. Results

### 2.1. Binding of Compounds in Cell Culture and to Zebrafish Proteins

A fair comparison of the potency of different compounds in any assay requires some knowledge of the amount of free unbound (Fu) compound that is available to exert a pharmacological response. We assessed the protein-binding of the three test compounds within zebrafish embryos and found a ca. 100-fold difference in the Fu-values (Table 1). Thus, infigratinib and vatalanib for the same concentration of compound were much more freely available than derazantinib within the zebrafish to exert their effect in this assay system. These differences in Fu-values were of a similar order to those determined for binding to plasma-protein in mice (Table 1).

### 2.2. DZB Affects Vascular Development In Vivo in a Dose-Dependent Manner

To assess the effects of DZB during vascular development, we added DZB to the water of zebrafish embryos at 6–8 h post-fertilisation (hpf), before the onset of endothelial cell specification. Our analysis of the transgenic zebrafish line *Tg(fli1a:EGFP)^y1^,* which marks specifically endothelial cells, showed that vascular development was perturbed upon adding increasing doses of DZB (Figure 1). In vehicle (dimethyl sulphoxide; DMSO)—treated embryos, intersegmental vessels (ISVs) branch out from the dorsal aorta (DA) at 22 hpf and migrate dorsally between the somites until they reach the dorsolateral roof of the neural tube, where tip cells from adjacent ISVs establish contacts and connect adjacent sprouts in a process termed anastomosis to form the dorsal longitudinal anastomotic vessel (DLAV) (Figure 1a,b). In DZB-treated embryos, sprouting angiogenesis was impaired at higher doses (0.3–10 μM) (Figure 1c–j and Appendix A).

Importantly, vascular disconnections between vessels at the level of the DLAV or between the DA and the intersegmental vessels were frequently observed at varying concentrations of DZB (0.1–3 μM). At these concentrations, the blood vessels also appeared thinner compared to control siblings (Figure 1g–j). At 10 μM, DZB appeared to be toxic since 70–80% of embryos between stages 20 and 45 hpf died, and thus in further studies we did not use DZB concentrations greater than 3 μM. Allowing for the protein-binding in zebrafish (Table 1), a concentration of 3 μM gave an Fu-value of 2 nM. This is a reasonable upper concentration to use, since the Fu-Cave (free unbound average concentration) for DZB in mouse plasma and subcutaneous xenografts is 0.6–1 nM and 3–5 nM, respectively, after daily treatment with an oral dose of 75–100 mg/kg DZB [10].

Taken together, our findings from in vivo time-lapse microscopy and in vitro embryonic lysate analyses support the finding that the range of concentrations of DZB which is able to exert an inhibitory effect on vascular developmental processes in zebrafish embryos is 0.1 to 3 μM. It is important to note that DZB treatment did not have any impact on vasculogenesis in zebrafish embryos, as shown by our observations in the dorsal aorta (DA) and the posterior cardinal vein (PCV), which formed normally, and appeared to have similar morphology to those of vehicle-treated embryos.

To further address the effects of DZB on blood vessel function, we analysed blood flow in the developing vascular system in *Tg(fli1a:EGFP)^y1^;Tg(gata1a:DsRed)^sd2^* embryos. We found that blood flow in larger vessels, i.e., within the dorsal aorta (DA) or the posterior cardinal vein (PCV), appeared normal in zebrafish embryos treated with DZB. However, due to vascular developmental defects in smaller vessels or capillaries, blood flow seemed to be perturbed in the ISVs and the DLAV and this phenotype matched the anastomosis defects described above. Nonetheless, even in those disrupted vascular networks, blood was able to flow in isolated segments that were connected to the main vessels (DA and PCV) and the blood flow pattern in intersegmental vessels appeared compartmentalized, following the pattern of the interconnected vascular network (Figure 2a–c and Appendix A).

### 2.3. Comparisons of DZB with the FGFR Inhibitor Infigratinib and VEGFR Inhibitor Vatalanib with Regard to Their Effects on Vascular Development

For these comparisons, infigratinib and vatalanib were used in the range of 0.1 to 0.3 μM and the zebrafish binding data (Table 1) showed that a nominal concentration of 0.1 μM gave Fu-values of 10 and 5 nM, respectively, which was in a similar range to the maximum concentration used for DZB (Fu = 2 nM). Both infigratinib and vatalanib increased embryo lethality at higher doses, with vatalanib being particularly toxic above a concentration of 1 μM (80–90% embryo lethality).

Our analysis on blood vessel function showed that treatment of zebrafish embryos with infigratinib led to mild impediment of blood flow, due to a discontinuous blood vessel network, similar to the effect seen with DZB (Figure 2d,d’). Vatalanib caused dramatic defects in blood vessel function, and blood flow was reduced even in the larger vessels (DA, PCV; Figure 2e,e’).

To further compare the effects of all three compounds on vascular development, we analysed sprouting angiogenesis and blood vessel anastomosis during embryonic development (Figure 3). Treatment with DZB induced mild defects in sprouting angiogenesis only at the higher concentrations (1–3 μM), when only a small portion of ISV sprouting was inhibited (Figure 1d–k and Figure 3h). Using the VEGFR inhibitor vatalanib, sprouting angiogenesis was dramatically reduced in all ISVs (Figure 3e,f,h), reminiscent of the effect of VEGFR inhibition in these vessels reported in previous studies [20,21]. Conversely, treatment with the FGFR inhibitor, infigratinib, did not have any effect on sprouting angiogenesis (Figure 3b,c,h). The requirement of VEGF/VEGFR signalling in sprouting angiogenesis is a hallmark of vascular development [1,16,22]. Our findings of the inhibition of angiogenic sprouting with DZB treatment support the notion that this compound interferes with VEGFR signalling during vascular development.

We further observed that DZB treatment resulted in a high number of disconnected vessels at intermediate and higher doses (0.1–3 μM) (Figure 1i,l and Figure 3d,g,i,j). Similarly, infigratinib treatment led to vascular disassembly at the DLAV and between ISVs and the dorsal aorta (Figure 3 b,c,a’,c’,i,j), suggesting that DZB induced these vascular defects by interfering with FGFR signalling. In vatalanib-treated embryos, however, it was not possible to analyse the effect of the compound during anastomosis of the DLAV, since the effect on angiogenic sprouting was so dramatic that the ISVs never reached the dorsolateral roof. These phenotypes, the anastomosis defects at the DLAV and blood vessel disconnections between the DA and ISVs, were not linked to arteriovenous fate, as shown by our analysis of the double transgenic line *Tg(fli1a:EGFP)^y1^;Tg(-0.8flt1:RFP)^hu5333^,* in which arterial endothelial cells are strongly labelled with RFP. Although DZB appeared to interfere at a higher rate with arterial vessel disconnections at the DLAV and venous vessel disconnections at the DA, both DZB and infigratinib had significant impact on vascular integrity of ISVs independent of their arterial or venous fate (Appendix A).

### 2.4. DZB Impairs Vascular Architecture

The observations on vascular integrity indicate that FGFR signalling is essential for some steps associated with blood vessel assembly. To further investigate the mechanism underlying vascular disconnections, we analysed the endothelial cell junctions using the double-transgenic line *Tg(fli1a:pecam1-EGFP)^ncv27^;Tg(kdrl:mCherry-CAAX)^s916^*, in which inter-endothelial cell junctions can be visualized by platelet endothelial molecule 1 (Pecam1-EGFP) and endothelial cell membranes are shown by mCherry-CAAX. In vehicle-treated embryos, ISV sprouting was accompanied by endothelial cell rearrangements, which occurred by extensive junctional remodeling, as previously reported [23,24]. As a consequence, ISVs appeared to have a pattern of highly elongated cell junctions (Figure 4a,a’). In contrast, treatment with DZB (0.3–3 μM) led to aberrant junctional patterns with shorter and linear junctions in a number of blood vessels (Figure 4b,c’and Appendix A) and this effect was dose-dependent. These junctional phenotypes coincided with blood vessels that appeared thinner in diameter with complete absence of lumen or presence of a smaller lumen compared to vehicle-treated embryos. Similar observations on junctional defects were made in embryos treated with infigratinib (Appendix A), indicating the necessity for FGFR signalling in junctional architecture and endothelial cell rearrangements.

### 2.5. DZB Impedes the Endothelial Cell Cycle

FGF signalling is a known promoter of cell proliferation during animal development and cancer [25,26]. As previously described, it coordinates cell cycle progression through controlling the expression of D-type cyclins, which are well-known cell-cycle regulators [27,28]. To assess the effect of the RTK inhibitors on cell cycle progression, we analysed cell divisions from time-lapse movies using the transgenic line *Tg(kdrl:EGFP-nls)^ubs1^*, in which the endothelial cell nuclei are visualized by EGFP-nls. Our analysis showed that DZB treatment (1–3 μM) inhibited cell divisions in ISV sprouts during the development of the vascular endothelium in zebrafish embryos (Figure 5a,c’’’,e, Appendix A). In a similar manner, the FGFR inhibitor infigratinib also inhibited cell cycle progression during sprouting of ISVs in concentrations ranging between 0.1 and 0.3 μM (Figure 5d,d’’’,e and Appendix A). These findings suggest that both DZB and infigratinib inhibit cell cycle progression potentially via FGFR signalling during vascular development.

The effects of all three compounds on various processes of vascular development in zebrafish embryogenesis are summarised in Table 2.

## 3. Discussion

### 3.1. Concentration Range Effect of DZB Reveals Dual Role as Anti-Angiogenic Drug

Exposure of zebrafish embryos using a concentration range of the RTK inhibitor DZB provides new insights regarding DZB as an anti-angiogenic drug. First, a wide range of DZB concentrations (0.1–3 μM) appears to induce defects in the developing vasculature of zebrafish embryos. Second, DZB affects distinct developmental processes and its effect is dependent on the concentration used. The higher concentrations (1–3 μM) of DZB, which are achievable in mouse plasma and tumours [10,29], led to strong sprouting defects in zebrafish embryos, a phenotype that identifies DZB as an angiogenesis inhibitor, similar to vatalanib. In contrast, lower DZB concentrations (0.03–0.3 μM) led to anastomosis defects, a phenotype that links DZB to blood vessel formation as well as to blood vessel maintenance, which was more similar to infigratinib. Third, these distinct defects caused by DZB appear to reflect the different signalling pathways of VEGFR and FGFR, which is supported by direct comparison to the effects produced by known kinase inhibitors that are specific for either VEGFR or FGFR [18,19]. Since it has already been demonstrated that DZB is efficacious against tumours with FGFR-aberrations both in preclinical in vivo tumour models [10] and in patients [11], the additional anti-angiogenic activity may increase its activity in such cancers in comparison to more specific FGFR-inhibitors such as infigratinib.

### 3.2. Implications for Anastomosis and FGFR Inhibition

As previously shown, FGFR1–3 are targeted by DZB [9,10]. In this report, we show that DZB has a detrimental impact on vascular morphogenesis and specifically on blood vessel anastomosis. This observation is consistent with a previous study, in which inhibition of vascular anastomosis in zebrafish was seen upon administration of a small molecule inhibitor specific for FGFRs [13]. A role for the FGFR pathway in anastomosis was also shown in a study, in which anastomosis between donor bioengineered vascular networks and host live vasculature from mouse was facilitated by FGF2 [30].

Apart from inhibiting anastomosis, DZB also interfered with cell junction architecture and blood vessel assembly. The endothelial cell junctional defects induced by DZB are reminiscent of the effect found in embryos treated with the FGFR inhibitor SSR and tightly links the observed junctional morphology and vascular integrity defects with previously characterised processes associated with FGF signaling [13,31]. 

In this study, we also showed that endothelial cell divisions were perturbed in DZB-treated zebrafish. Cell cycle regulation is another hallmark of the FGF signalling system [27]. Taken together, DZB treatment appears to affect many key morphogenetic processes linked to FGF signalling in vascular development in vivo.

### 3.3. Implications for Sprouting

Anastomosis and blood vessel integrity were affected at low concentrations of DZB (0.1– 0.3 μM), and an anti-angiogenic effect was clearly detectable at the higher concentrations (1–3 μM). These higher concentrations are achievable in mice at tolerable oral doses of 75–100 mg/kg, qd [10], when taking into account the different binding of DZB to zebrafish protein and proteins in plasma and tumors. A connection between DZB and VEGFR-signalling is evidenced by the effect that DZB has on sprouting angiogenesis in zebrafish, a process which is tightly linked to VEGF/VEGR signaling [15,16]. Overall, the observed phenotypes are comparable to those of vatalanib, a potent VEGFR inhibitor. This observation, together with the fact that VEGFR2 has been shown to be a kinase target of DZB both in vitro and in vivo in mice and subcutaneous xenografts [9,28], indicates that DZB acts on angiogenic sprouting via perturbation of VEGFR signalling. 

This study provides novel evidence for an in vivo effect of the inhibitor DZB on vascular morphogenesis and opens the possibility for use of this compound as an anti-angiogenic drug by exploiting the fact that it interferes with both FGFR and VEGFR signalling in very distinct ways, as reflected by its activity on vascular angiogenesis.

## 4. Materials and Methods

### 4.1. Measurement of Compound Binding to Mouse Plasma Protein and Zebrafish Protein

Protein binding was measured by rapid equilibrium dialysis (ThermoScientific, Reusable RED Plate) in zebrafish embryos. Zebrafish embryos (0.48 g) were first homogenized with 9.6 mL of MilliQ water with a FastPrep-24 5G (MP Bio, TallPrep Lysing Matrix Z), using a ratio of 1:20 (v:v) eq. The equilibrium between zebrafish homogenate and buffer was assumed to be achieved after 20 h incubation at ≈320 rpm for all the three compounds. The zebrafish homogenate was spiked at 0.1, 0.3 and 1 μM with the test compound (i.e., 4.04 μL of, respectively, 10, 30 or 100 μM DMSO working solution added into 400 μL of homogenate_1% final DMSO concentration). Then, 100 μL of zebrafish homogenate was placed into the sample (donor) chamber, while 300 μL of phosphate buffered saline (PBS) was added to the buffer (acceptor) chamber. After 20 h mixing with an orbital shaker (≈320 rpm) at 37 °C, 40 μL aliquots were pipetted from both fish homogenate and buffer chambers. A corresponding 40 μL of homogenate was added to the buffer sample and an equal volume of buffer to the collected homogenate. Next, 240 μL of the quenching solution (acetonitrile (ACN) containing 0.5 μg/mL of BAL087-d4 as internal standard) was added to the homogenate:buffer samples. The samples were mixed and then centrifuged for 20 min at 4000 rpm. The supernatants were transferred into new vials and injected into the LC-MS/MS system. Each experiment was performed in triplicate.

### 4.2. Zebrafish Maintenance and Transgenic Lines/Strains 

Zebrafish (*Danio rerio*) were maintained in standard conditions [32] and experimental procedures involving zebrafish embryos were carried out at the Biozentrum/Universität Basel in accordance to Swiss federal guidelines and were approved by the Kantonales Veterinäramt of Kanton Basel-Stadt. Zebrafish lines were maintained under licenses 1014H and 1014G1 issued by the Veterinäramt-Basel-Stadt. Embryos were staged according to [33] hours post-fertilization (hpf) at 28.5 °C. The following zebrafish lines were used: *Tg(fli1a:EGFP)^y1^* [34] (labels all endothelial cells), *Tg(gata1a:DsRed)^sd2^* [35] (labels all erythrocytes), *Tg(kdrl:EGFP-nls*)^ubs1^ [36] (labels endothelial cell nuclei), *Tg(kdrl:mCherry-CAAX)^s916^* [37] (labels endothelial cell membranes), *Tg(fli1a:pecam1-EGFP)^ncv27^* [38] (labels endothelial cell junctions) and *Tg(-0.8flt1:RFP)^hu5333^* [39] (strongly labels arterial endothelial cells).

### 4.3. In Vivo Time-Lapse Microscopy and Image Analysis

Staged embryos were anaesthetised with 0.08% tricaine (Sigma, MS-222) solution and subsequently mounted on a 4-well 35-mm glass bottom petri dish (35/10 mm, CELLview™) with 0.7% low-melting agarose (Sigma) containing 0.08% tricaine and 0.003% 1-phenyl 2-thiourea (PTU, Sigma). Time-lapse imaging was acquired with a Zeiss880 confocal microscope using a 25x (NA = 0.8) oil immersion objective and Z-stacks with a 0.7 to 1 µm step size were acquired every 10–20 min. ImageJ software (http://fiji.sc/) was used for image and video analysis and OMERO software (https://www.openmicroscopy.org/omero/) was used for data storage and figure preparation. 

### 4.4. Pharmacological Treatment

DMSO stock solution (10 mM) of RTK inhibitor DZB (BAL087, Basilea) was diluted in both agarose and fish/egg water covering imaged embryos to the final concentration specified in the results (0.001–10 μM). Similarly, DMSO solutions (100 μM) of infigratinib (BGJ398) or vatalanib (PTK/ZK) were diluted in fish water to achieve concentrations of 0.1–0.3 μM. The compounds were added to embryo fish water E3 (5 mM NaCl, 0.17 mM KCl, 0.33 mM CaCl2, 0.33 mM MgSO4, pH 7.0) at 8–10 hpf onwards, before the onset of endothelial cell specification, as previously described [13,33,40].

### 4.5. Statistical Analysis and Graphs

Data were analysed using GraphPad Prism 8.0. software and were presented as mean  ±  SEM (error bars). Statistical analysis was performed by the non-parametric Mann−Whitney’s test or the parametric Student’s *t* test. Significance was considered when *p* values were lower than 0.05, ns indicated non- significance, while * *p* < 0.05; ** *p*  <  0.01; *** *p*  <  0.001 and **** *p*  <  0.001 were considered statistically significant. Sample size and experimental replicates are indicated in the Figure legends.

## Figures and Tables

**Figure 1 pharmaceuticals-14-00025-f001:**
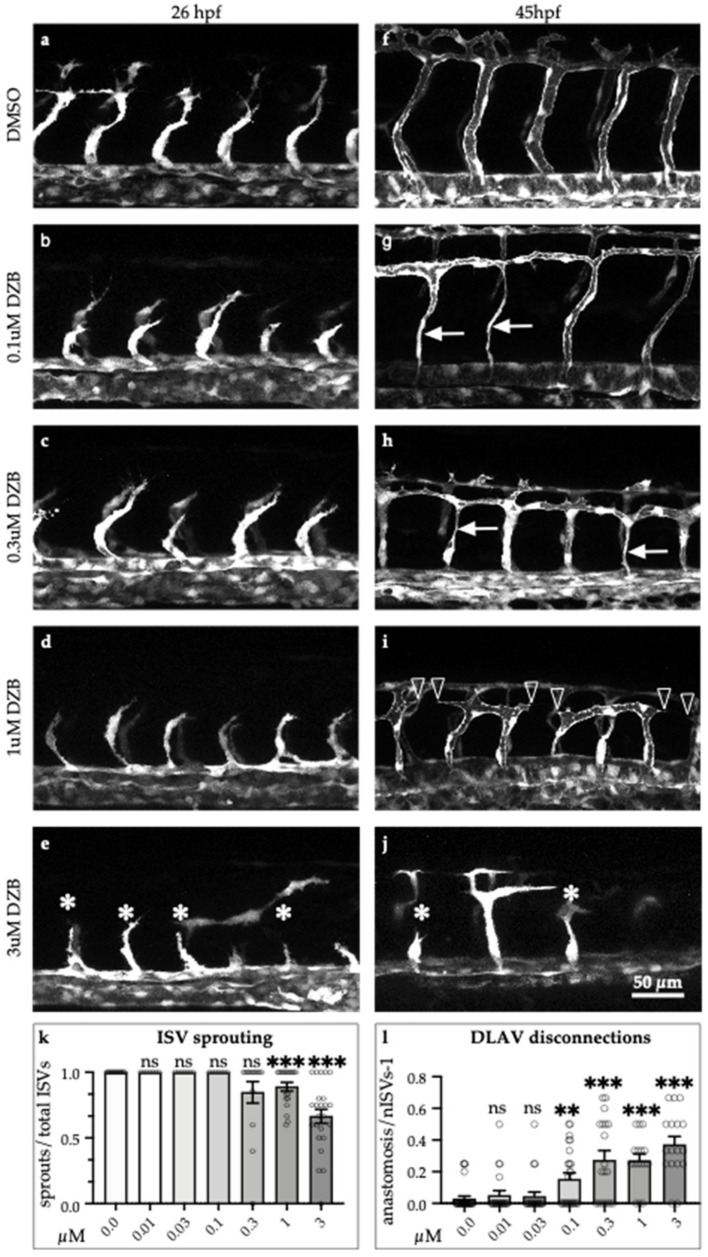
Derazantinib inhibits vascular development in vivo in a dose-dependent manner. Confocal images of GFP+ blood vessels in the trunk of *Tg(fli1:EGFP)^y1^* zebrafish embryos at 26 hpf (**a**–**e**) or 45 hpf (**f**–**j**) after exposure to DMSO as vehicle control (**a** and **f**) or increasing concentrations of DZB in the swimming water (**b**–**e** and **g**–**j**). Blood vessel development was disrupted using concentrations between 0.1 and 3 µM DZB. (**k**) Quantitative analysis of ISV sprouts that had reached the top roof and started to form the DLAV were normalised to the total number of ISVs per embryo (*n* ≥ 15 embryos per treatment were analysed from three independent experiments), embryos were treated using increasing concentrations of DZB. (**l**) Quantitative analysis of ISV sprouts that are disconnected at the DLAV were normalised to the total number of ISVs-1 (total number of connections) per embryo (*n* ≥ 15 embryos per treatment were analysed from three independent experiments) and embryos were treated using increasing concentrations of DZB. Data in **k****,****l** represent mean ± S.E.M. (error bars), ns: not significant, ** *p* < 0.01, *** *p* < 0.001. Arrows indicate thinner blood vessels, arrowheads show disconnected blood vessels and asterisks mark sprouting defects. Scale bar, 50 µm. ISV, intersegmental vessel; DLAV, dorsal longitudinal anastomotic vessel; DA, dorsal aorta; PCV, posterior cardinal vein. See also Appendix A.

**Figure 2 pharmaceuticals-14-00025-f002:**
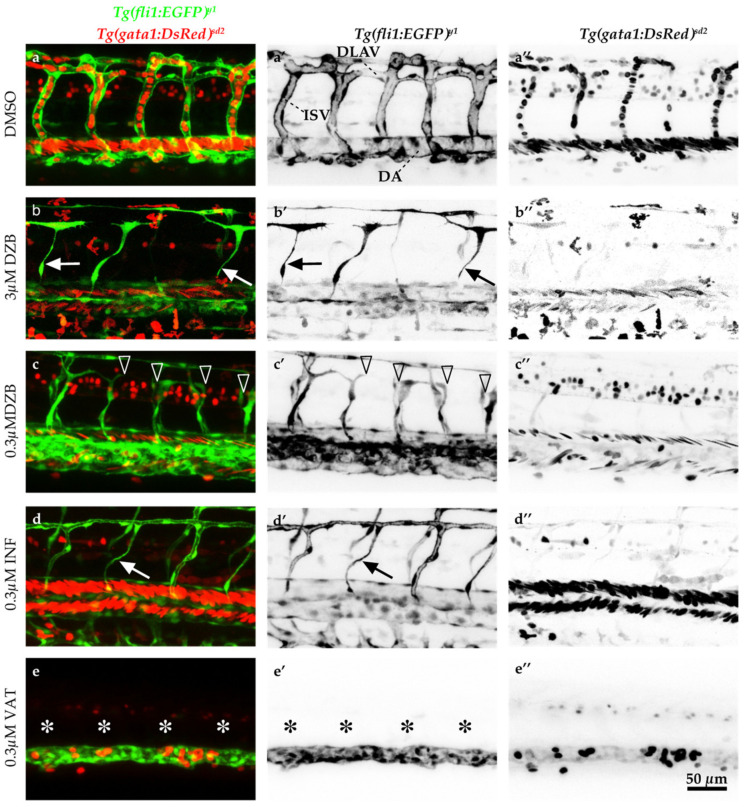
Comparisons of derazantinib, infigratinib and vatalanib blood vessel function. Confocal images of GFP+ blood vessels and DsRed+ erythrocytes in the trunk of *Tg(fli1:EGFP)^y1^/Tg(gata1:DsRed^)sd2^* zebrafish embryos at 45 hpf after exposure to DMSO as control (**a**), derazantinib (DZB; **b**,**c**), infigratinib (INF; **d**) or vatalanib (VAT; **e**) in the swimming water. Panels **a’**–**e’** depict only the GFP channel and panels **a’’**–**e’’** depict only the DsRed channel. Although blood flow appeared in DZB and INF treatments at the dorsal aorta (DA), blood circulation was inhibited at the intersegmental vessels (ISVs) and dorsal longitudinal anastomotic vessel (DLAV) due to disruption of the vascular network (arrowheads) or due to thinner blood vessels (arrows) using DZB or INF treatments. Blood flow and blood vessel sprouting (asterisks) were disrupted in VAT treatment. Scale bar, 50 µm. See also Appendix A.

**Figure 3 pharmaceuticals-14-00025-f003:**
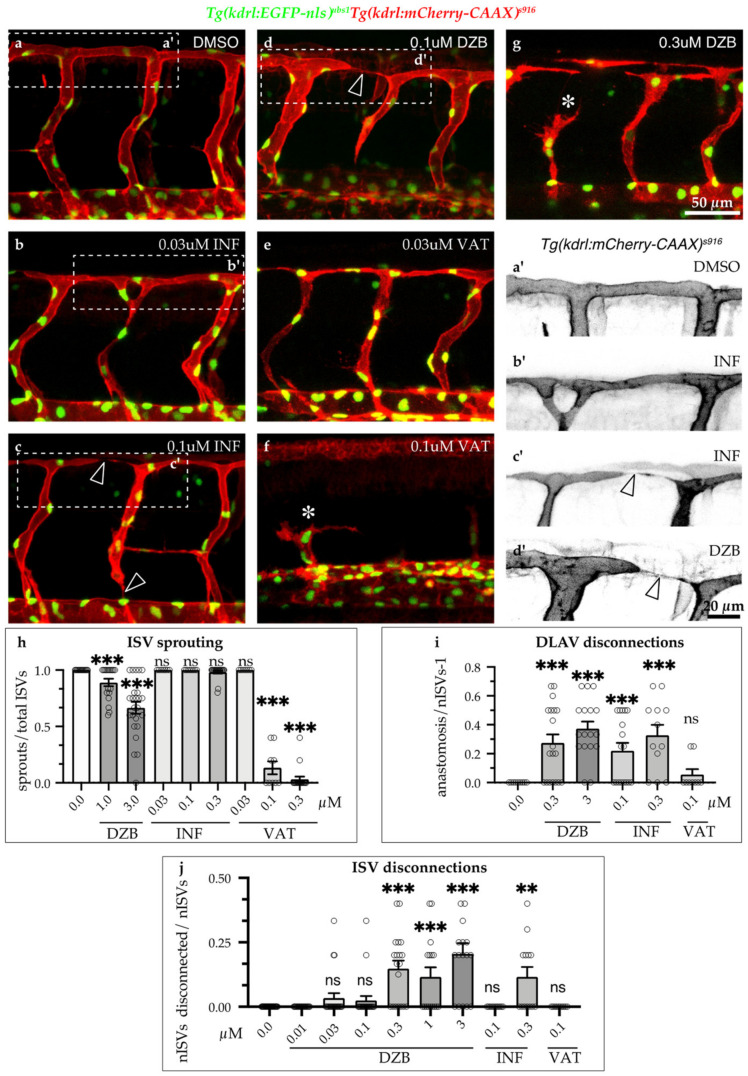
Comparisons of derazantinib, infigratinib and vatalanib during vascular development. Confocal images of GFP+ endothelial cell nuclei and mCherry+ endothelial cell membranes in *Tg*(*kdrl:EGFP-nls*)*^ubs1^*/*Tg*(*kdrl:mCherry-CAAX*)*^s916^* transgenic embryos from 45 hpf in vehicle (DMSO, **a**) or embryos treated with INF (**b,c**), VAT (**e,f**) or DZB (**d,g**). Treatment with DZB or INF led to blood vessel disconnections (arrowheads) compared to control (**a**–**d**). Panels **a**’–**d**’ depict zoom-in images of the outlined boxes in a–d marking blood vessel anastomosis defects (arrowheads), scale bar 20 µm. Treatment with DZB and VAT led to sprouting defects (asterisks; **e**–**g**). Scale bar for **a**–**g**, 50 µm. (**h**) Quantitative analysis of ISV sprouts that had reached the top roof and started to form the DLAV were normalised to the total number of ISVs per embryo (*n* ≥ 15 embryos per treatment were analysed from three independent experiments). (**i**) Quantitative analysis of ISV sprouts that were disconnected at the DLAV were normalised to the total number of ISVs-1 (total number of connections) per embryo (*n* ≥ 15 embryos per treatment were analysed from three independent experiments). (**j**) Quantitative analysis of ISV sprouts that are disconnected from the dorsal aorta were normalised to the total number of ISVs per embryo (total number of connections) per embryo (*n* ≥ 15 embryos per treatment were analysed from three independent experiments). Data in **h**–**j** represent mean ± S.E.M. (error bars), ns: not significant, ** *p* < 0.01, *** *p* < 0.001. Statistical analysis was performed with the two-sided Mann–Whitney test. See also Appendix A.

**Figure 4 pharmaceuticals-14-00025-f004:**
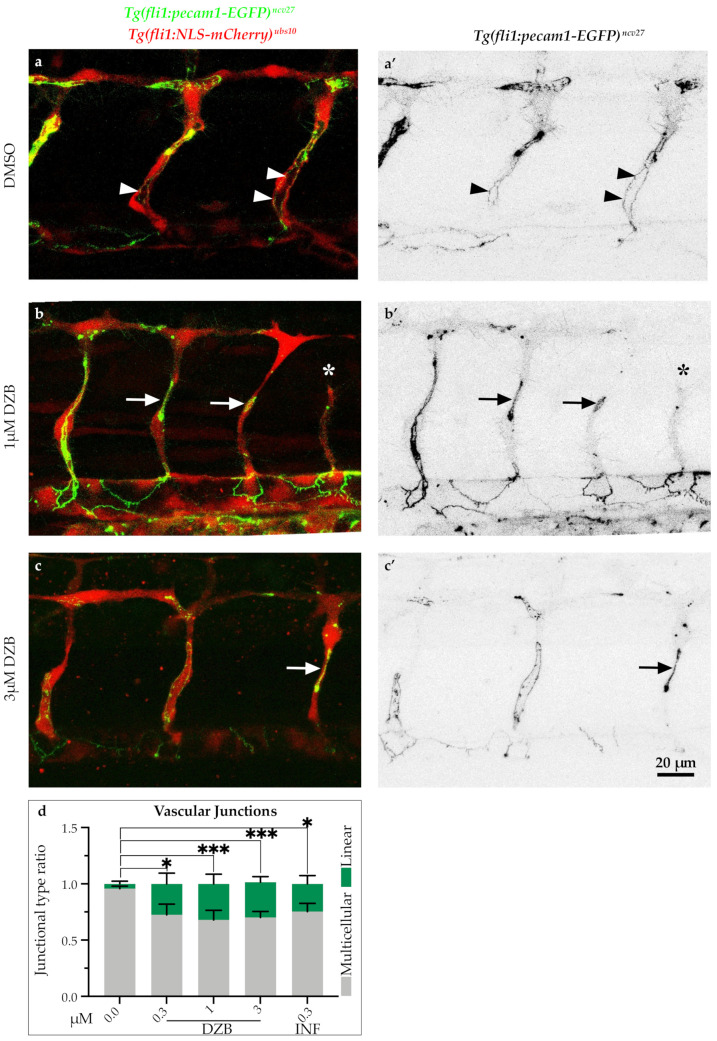
Derazantinib interferes with endothelial cell junctions. Confocal images of GFP+ endothelial cell junctions and mCherry+ endothelial cell nuclei in *Tg(fli1:pecam1-EGFP)^ncv27^*/*Tg(fli1:NLS-mCherry)^ubs10^* transgenic embryos from 38 hpf after treatment with DMSO (control, **a**) or DZB (**b** and **c**). **a’**–**c’** depict only the GFP+ channel. Arrowheads point to multicellular continuous cell junctions, indicative in the control group (DMSO, **a** and **a’**), while arrows point to linear discontinuous junctions, indicative in the DZB-treated embryos (**b** and **c’**). The asterisk marks a sprouting defect (**b** and **b’**). (**d**) Quantification of ratio of continuous cell junctions to linear discontinuous junctions in ISVs per embryo (*n* ≥ 15 embryos per treatment were analysed from three independent experiments). Data in d represent mean ± S.E.M. (error bars), * *p* < 0.05, *** *p* < 0.001. Statistical analysis was performed with the two-sided Mann–Whitney test. Scale bar, 20 µm. See also Appendix A.

**Figure 5 pharmaceuticals-14-00025-f005:**
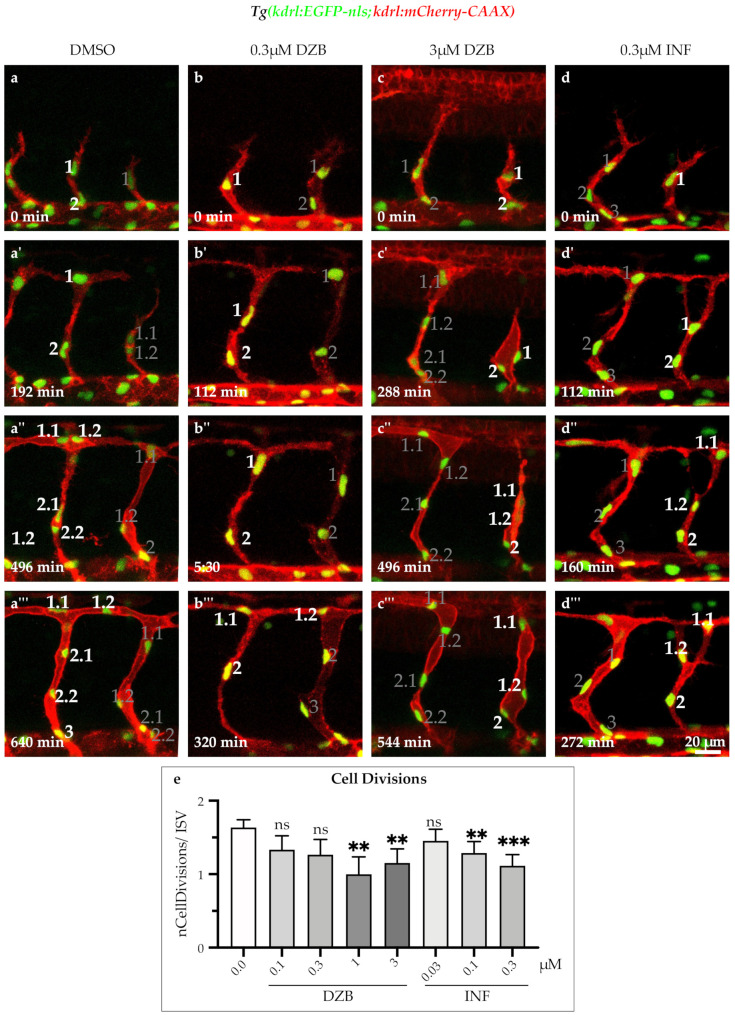
Derazantinib and infigratinib inhibit endothelial cell cycle. Time-lapse images of sprouting ISVs of GFP+ endothelial cell nuclei and mCherry+ endothelial cell membranes in *Tg*(*kdrl:EGFP-nls*)*^ubs1^*/*Tg*(*kdrl:mCherry-CAAX*)*^s916^* transgenic embryos from 26 hpf after treatment with DMSO (**a**–**a’’’**), 0.3 uM DZB (**b**–**b’’’**), 3 uM DZB (**c**-**c’’’**) or 0.3 uM INF(**d**–**d’’’**). Numbers indicate cell nuclei (i.e., 1, 2, 3,...) or cell nuclei arising after cell division (i.e., 1.1, 1.2,...). Number of mitotic endothelial cells was reduced in DZB-treated (**c**–**c’’’**) or INF-treated embryos (**d**–**d’’’**). (**e**) Quantitative analysis of mitotic endothelial cells in ISVs per embryo (*n* ≥ 10 embryos per treatment were analysed from three independent experiments). Data in e represent mean ± S.E.M. (error bars), ns: not significant, ** *p* < 0.01, *** *p* < 0.001. Statistical analysis was performed with the two-sided Mann–Whitney test. Scale bar, 20 µm. See also Appendix A.

**Table 1 pharmaceuticals-14-00025-t001:** Analysis of % of bound and unbound compounds in embryonic protein lysates.

Compound	%PB-ZF	%Fu-ZF ^1^	Ratio in ZF (Fu)	%PPB-Mouse	%Fu-PPB	Ratio in Mice (Fu)
derazantinib	99.93	0.067 ± 0.003	1	99.96	0.05 ± 0.01	1
infigratinib	95.03	4.97 ± 0.82	71	98.7	1.3 ± 0.00	29
vatalanib	90.05	9.95 ± 1.31	142	95.87	4.1 ± 0.18	92

^1^ Binding was determined at three different concentrations (0.1, 0.3 and 1.0 μM) for the compounds as described in methods. Results show the mean ± SEM value (*n* = 3) for the respective Fu-compound at a concentration of 1 μM (binding was not found to be affected by the compound concentration; results not shown). PB: Protein-binding, Fu: free unbound, ZF: zebrafish, PPB: plasma protein binding.

**Table 2 pharmaceuticals-14-00025-t002:** Effects on vascular development and vascular features/functions using the compounds.

Developmental Process	Derazantinib	Derazantinib	Infigratinib	Vatalanib
Concentration range ^1^	0.1–0.3 μM	1–3 μM	0.1–0.3 μM	0.1–0.3 μM
Calculated Fu in zebrafish ^2^	0.07–0.2 nM	0.7–2 nM	5–15 nM	10–30 nM
ISV sprouting angiogenesis	No/mild defects	Strong defects	No defects	Strong defects
Anastomosis	Mild defects	Moderate/strong defects	Moderate/strong defects	n.a.
ISV-Aorta Connections	Mild defects	Moderate defects	Moderate defects	No defects
Lumenization	Mild defects	Mild defects	Mild defects	Moderate defects
Blood flow	No defects	Mild defects	Mild defects	Moderate defects
Cell junctions	Mild defects	Moderate/strong defects	Mild defects	n.a.
Cell cycle	Mild defects	Moderate defects	Moderate defects	n.a.

^1^ various concentrations ranges and ^2^ the corresponding calculated free-unbound (Fu) amount of the compounds from zebrafish embryo lysates.

## Data Availability

Data available on request.

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
