# Peer review of "Probing the Effects of the FGFR-Inhibitor Derazantinib on Vascular Development in Zebrafish Embryos"

_pharmaceuticals, 2020, doi:10.3390/ph14010025_

Round 1

Reviewer 1 Report

The manuscript, entitled “Probing the effects of the FGFR-inhibitor Derazantinib on vascular development in zebrafish embryos”, investigates the in vivo effect of FGFR1-3 inhibitor Derazantinib (DZB) on vascular morphpgenesis as an anti-angiogenic drug in the vascular development of zebrafish. Mechanistically, low concentrations of DZB led to anastomosis, endothelial cell junction defects and cell cycle inhibition through FGF signaling system, while higher concentration of DZB led to strong sprouting defects in zebrafish embryos via perturbation of VEGFR signaling. This study reveals a potential dual role for DZB as a potent anti-angiogenic treatment and the results are solid and well organized. However, specific issues suggested for attention include:

Major issues:

  1. In this study, higher concentration of DZB led to strong sprouting defects in vascular development of zebrafish, a phenotype similar to vatalanib. As a potent anti-angiogenic treatment for cancer therapy, what is DZB’s advantage compares with vatalanib? Clarify the advantages of DZB would enhance the practicality of the article.

Minor issue:

  1. Figure 1k: x-axis contents and column diagrams did not match.
  2. The arrows in the figures should be consistent.
  3. Figure 1 legend: without arrowheads’ illustration.
  4. Asterisks did not show in Figure 2.
  5. Please check the figures and figure legends carefully.

Author Response

Response to Major Issues:

We added the following (Line 297):

“Since it has already been demonstrated that DZB is efficacious against tumours with FGFR-aberrations both in preclinical in vivo tumour models [10] and in patients [11], the additional anti-angiogenic activity may increase its activity in such cancers in comparison to more specific FGFR-inhibitors such as infigratinib”.

Thus, we do not necessarily envisage DZB to be an alternative to specific VEGFR-inhibitors, but that it may have advantages compared to specific FGFR-inhibitors.

In addition, we have added a sentence about the clinical progress of DZB to the Introduction (Line-42):

“DZB has shown significant efficacy in patients with cholangiocarcinoma in a phase-1/2 trial [11] and further trials have initiated in both urothelial (NCT04045613) and gastric cancer (NCT04604132).

Response to Minor Issues:

 Response to issue 1: We exchanged the graph 1k to include the treatment 0.01 uM DZB which was missing. Now as all the contents and columns match (Lines 103-105)

Response to issues 2-4: We have exchanged all the figures with new ones, so that the arrows, arrowheads and asterisks are consistent and formatted the font to match the Manuscript one.

Response to issues 5: We corrected the figure legends to match the new Figures.

Furthermore, we also edited the following to be consistent within the MS:

  1. We included text and an additional reference at the Methods (Line 354) which was missing.
  1. We exchanged the word Supplementary Figure# for Figure S#. (Lines 86, 119, 187,222,225, 251, 253)
  1. We exchanged the word(s) Supplementary Movies or Movies for Videos (Lines 135, 147, 251, 271)

Reviewer 2 Report

Kotini et al have undertaken a study to investigate the effects of the inhibitor DZB on FGF and VEGF signalling comparing to other known specific inhibitors of these two signalling pathways. 

Do the authors have any indication in if there is a particular receptor, FGF or VEGF, that are mediating the effects that are seen following the inclusion of the inhibitors? VEGFR2 has been suggested to be key signalling receptor for VEGF and there are differences in expression of FGF receptors in the zebrafish with age. 

DZB is also suggested to inhibit CSF1R that can regulate macrophage activity. Did the authors look at effects on macrophages following addition of DZB to water since macrophages are also implicated in angiogenesis being found at areas of sprouting? If so, are there reduced macrophage numbers that could be linked to the various defects observed using the three inhibitors. 

The manuscript is well written and it is appreciated that there is a table at the end that describes all the changes observed using the three different inhibitors to allow the reader easier understanding for comparison of the changes in signalling pathways. 

Author Response

Response to Issue 1:  

We have demonstrated a dose-dependent effect on VEGFR2 in cells in vitro [Ref-28].

Response to Issue 2:  

We have demonstrated a dose-dependent effect on CSF1R in freshly isolated mouse macrophages ex vivo [Ref-10].

Furthermore, we also edited the following to be consistent within the MS:

  1. We included text and an additional reference at the Methods (Line 354) which was missing.
  1. We exchanged the word Supplementary Figure# for Figure S#. (Lines 86, 119, 187,222,225, 251, 253) 
  1. We exchanged the word(s) Supplementary Movies or Movies for Videos (Lines 135, 147, 251, 271)